# The Neutrophil-to-Monocyte Ratio and Lymphocyte-to-Neutrophil Ratio at Admission Predict In-Hospital Mortality in Mexican Patients with Severe SARS-CoV-2 Infection (Covid-19)

**DOI:** 10.3390/microorganisms8101560

**Published:** 2020-10-10

**Authors:** Salma A. Rizo-Téllez, Lucia A. Méndez-García, Cruz Flores-Rebollo, Fernando Alba-Flores, Raúl Alcántara-Suárez, Aarón N. Manjarrez-Reyna, Neyla Baltazar-López, Verónica A. Hernández-Guzmán, José I. León-Pedroza, Rogelio Zapata-Arenas, Antonio González-Chávez, Joselín Hernández-Ruíz, José D. Carrillo-Ruíz, Raúl Serrano-Loyola, Guadalupe M. L. Guerrero-Avendaño, Galileo Escobedo

**Affiliations:** 1Laboratory of Immunometabolism, Research Division, General Hospital of Mexico “Dr. Eduardo Liceaga”, Mexico City 06720, Mexico; sart.17.04@gmail.com (S.A.R.-T.); angelica.mendez.86@hotmail.com (L.A.M.-G.); cruzifr3@gmail.com (C.F.-R.); feralflo.mcp@gmail.com (F.A.-F.); raul_as02@hotmail.com (R.A.-S.); aaron.manjarrez@gmail.com (A.N.M.-R.); 2PECEM, Facultad de Medicina, Universidad Nacional Autónoma de México, Coyoacán, Mexico City 04510, Mexico; 3Research Coordination at Central Laboratories, General Hospital of Mexico “Dr. Eduardo Liceaga”, Mexico City 06720, Mexico; neylabaltazar@yahoo.com.mx (N.B.-L.); veronicahgm@gmail.com (V.A.H.-G.); 4Department of Intensive Medical Therapy, General Hospital of Mexico “Dr. Eduardo Liceaga”, Mexico City 06720, Mexico; jisrael_leon@hotmail.com; 5Department of Internal Medicine, General Hospital of Mexico “Dr. Eduardo Liceaga”, Mexico City 06720, Mexico; el_zarmx@hotmail.com (R.Z.-A.); antglez51@yahoo.com.mx (A.G.-C.); 6Clinical Pharmacology Unit, General Hospital of Mexico “Dr. Eduardo Liceaga”, Mexico City 06720, Mexico; hernandezjoselin@hotmail.com; 7Department of Neurology and Neurosurgery, General Hospital of Mexico “Dr. Eduardo Liceaga”, Mexico City 06720, Mexico; josecarrilloruiz@yahoo.com; 8Department of Radiology and Imaging, General Hospital of Mexico “Dr. Eduardo Liceaga”, Mexico City 06720, Mexico; serranoraul48@yahoo.com.mx (R.S.-L.); gpeguerrero@prodigy.net.mx (G.M.L.G.-A.)

**Keywords:** SARS-Cov-2, Covid-19, mortality predictor, in-hospital death risk, lymphocyte, monocyte, neutrophil, intensive care unit

## Abstract

There is a deep need for mortality predictors that allow clinicians to quickly triage patients with severe coronavirus disease 2019 (Covid-19) into intensive care units at the time of hospital admission. Thus, we examined the efficacy of the lymphocyte-to-neutrophil ratio (LNR) and neutrophil-to-monocyte ratio (NMR) as predictors of in-hospital death at admission in patients with severe Covid-19. A total of 54 Mexican adult patients with Covid-19 that met hospitalization criteria were retrospectively enrolled, followed-up daily until hospital discharge or death, and then assigned to survival or non-survival groups. Clinical, demographic, and laboratory parameters were recorded at admission. A total of 20 patients with severe Covid-19 died, and 75% of them were men older than 62.90 ± 14.18 years on average. Type 2 diabetes, hypertension, and coronary heart disease were more prevalent in non-survivors. As compared to survivors, LNR was significantly fourfold decreased while NMR was twofold increased. LNR ≤ 0.088 predicted in-hospital mortality with a sensitivity of 85.00% and a specificity of 74.19%. NMR ≥ 17.75 was a better independent risk factor for mortality with a sensitivity of 89.47% and a specificity of 80.00%. This study demonstrates for the first time that NMR and LNR are accurate predictors of in-hospital mortality at admission in patients with severe Covid-19.

## 1. Introduction

The outbreak of coronavirus disease 2019 (Covid-19) is an ongoing global pandemic caused by the novel severe acute respiratory syndrome coronavirus-2 (SARS-CoV-2) that has affected the lives of millions of people worldwide [1]. Although more than 85% of patients with Covid-19 show a self-limiting illness with symptoms such as mild fever, dry cough, and fatigue, some patients develop severe pneumonia that can progress to acute respiratory distress syndrome, multiple organ failure, and death [2]. Mortality rates directly attributed to severe Covid-19 ranges from 3% in countries like China to 10% in countries like Spain [3,4]. However, mortality rates above 10% have been alarmingly reported in other countries like Italy, UK, and Mexico [4,5,6]. In particular, a 145% excess mortality associated with severe Covid-19 has been recently estimated for some Mexico’s cities, which is presumably associated with higher prevalence of comorbidities such as obesity, type 2 diabetes (T2D), and hypertension in our population [7]. Thus, it is of great importance to develop novel tools that help us to estimate the mortality risk at admission in patients with severe Covid-19 that need hospitalization, with the aim of quickly triaging them into intensive care units.

Clinical features of patients with Covid-19 have revealed a number of potential biochemical markers associated with in-hospital mortality. In particular, the blood levels of D-dimer, ferritin, C-reactive protein (CRP), troponin I, lactate dehydrogenase (LDH), and procalcitonin have been extensively studied due to their apparently good accuracy to discriminate patients with the most severe courses of the disease [8,9,10,11,12,13]. In parallel, numerous studies have also proposed the use of hematological markers such as lymphocyte, monocyte, and neutrophil counts that seem to associate with increased severity and mortality in patients with Covid-19 [14,15]. Actually, it has been consistently demonstrated that, in seriously ill patients with Covid-19, the blood count of lymphocyte and monocyte populations decreases whereas neutrophil counts increase [15]. However, total counts of these white blood cells (WBC) are merely associated with severity and mortality of Covid-19 but fails to predict death risk in clinical practice [14]. For this reason, a growing body of evidence has now proposed that ratios among lymphocytes, monocytes, and neutrophils are more accurate to predict mortality than cell count of lymphocytes, monocytes, and neutrophils by itself [16,17]. As a matter of fact, numerous reports have previously shown that the lymphocyte-to-neutrophil ratio (LNR) and the neutrophil-to-monocyte ratio (NMR) are markers associated with poorer survival in acute lung bacterial infection, sepsis, and many solid tumors including gastroesophageal, colorectal, pancreatic, prostate, and breast carcinoma [18,19,20]. Nevertheless, NLR and NMR have been scarcely explored as mortality predictors in patients with severe Covid-19, especially in countries with a disproportionate number of deaths related to this disease, such as Mexico.

For this reason, our main goal was to assess the efficacy of LNR and NMR as predictors of mortality at admission in adult patients with severe SARS-CoV-2 infection that met hospitalization criteria.

## 2. Material and Methods

### 2.1. Subjects

A total of 54 Mexican patients of both sexes aged 18 years or older, with severe Covid-19, admitted to the Department of Intensive Medical Therapy of the General Hospital of Mexico, were retrospectively enrolled in the study from 23 March 2020 to 26 June 2020. The diagnosis of Covid-19 was confirmed by specific detection of the SARS-CoV-2 in nasopharyngeal swabs using quantitative polymerase chain reaction (qPCR) according to the World Health Organization (WHO) technical guidance [21]. Patients seriously ill with Covid-19 that needed hospitalization were enrolled in the study if they met at least one of the following criteria: oxygen saturation level (SpO_2_) < 93% on room air, respiratory distress > 30 breaths/min, and/or > 50% lung involvement on imaging. Patients were excluded from the study if they had previous diagnosis of cancer, end-stage kidney and/or liver failure, endocrine disorders, infectious diseases, and/or autoimmune disease. We also excluded from the study human immunodeficiency virus (HIV), hepatitis C virus (HCV), and hepatitis B virus (HBV)-seropositive individuals, patients under long-term immunomodulatory medication including non-steroidal anti-inflammatory drugs, unconscious patients that had lost ability to respond, and pregnant or lactating women. All participants provided written informed consent, previously approved by the institutional ethical committee of the General Hospital of Mexico (registration number of the ethical code approval: DI/20/501/03/17), which guaranteed that the study was conducted in rigorous adherence to the principles that were described in the 1964 Declaration of Helsinki and its posterior amendment in 2013.

### 2.2. Data Collection

Laboratory and clinical data were collected at hospital admission from all patients enrolled in the study using the digital version of the electronic health record of the General Hospital of Mexico. Demographic data and medical history including previous diagnosis of obesity (body mass index (BMI) > 30 kg/m^2^), T2D (glycated hemoglobin (HbA1c) > 6.5% or medication to treat T2D), high blood pressure (>130/80 mm Hg or medication to treat hypertension), and coronary heart disease (CHD) were personally recorded at the Department of Intensive Medical Therapy. Furthermore, the need for invasive mechanical ventilation (IMV) with an oral endotracheal tube attached to a mechanical ventilator, time to extubation, inpatient days up to the main outcome, and drug regimen to treat Covid-19 were personally recorded at the Department of Intensive Medical Therapy. All patients were followed-up with daily, until hospital discharge or death, and then were assigned to survival or non-survival groups, respectively.

### 2.3. Laboratory Measures

Laboratory measures included biochemical parameters, liver function tests, and kidney function test as follows: blood glucose, triglycerides, total cholesterol, low-density lipoproteins (LDL), high-density lipoproteins (HDL), alanine aminotransferase (ALT), aspartate aminotransferase (AST), gamma glutamyl transferase (GGT), alkaline phosphatase (ALP), total bilirubin, direct bilirubin, indirect bilirubin, amylase, lipase, creatinine phosphokinase (CPK), creatinine kinase myocardial band (CK-MB), total protein, albumin, LDH, serum creatinine, uric acid, calcium, phosphorus, chloride, potassium, sodium, and magnesium. Additional measures also included hematological and immune parameters, and coagulation tests as follows: WBC count, red blood cell (RBC) count, lymphocytes, monocytes, eosinophils, basophils, band cells, neutrophils, hemoglobin, hematocrit, mean corpuscular volume (MCV), mean corpuscular hemoglobin (MCH), red blood cell distribution width (RDW), platelet count, mean platelet volume, CRP, troponin I, ferritin, procalcitonin, myoglobine, D-dimer, fibrinogen, prothrombin time (PT), international normalized ratio (INR), activated partial thromboplastin time (aPTT), and thrombin time (TT). LNR resulted of dividing the total lymphocyte count by the total neutrophil count. NMR resulted of dividing the total neutrophil count by the total monocyte count. All laboratory parameters were measured using the Beckman Coulter DxC 700 AU Chemistry Analyzer (Beckman Coulter Inc., Brea, CA, USA), the Coulter LH 780 Hematology Analyzer (Beckman Coulter Inc., Brea, CA, USA), and the BCS^®^ XP System (Siemens Healthcare GmbH, Erlangen, Germany) following strict adherence to the standard operating procedures.

### 2.4. Statistics

All laboratory parameters were recorded at hospital admission and then analyzed retrospectively. Normality of data distribution was estimated by the Shapiro–Wilk test. The unpaired Student’s T-test was used to compare age, BMI, blood glucose, triglycerides, total cholesterol, LDL, HDL, ALT, AST, GGT, alkaline phosphatase, total bilirubin, direct bilirubin, indirect bilirubin, total protein, albumin, LDH, serum creatinine, uric acid, calcium, phosphorus, chloride, potassium, sodium, magnesium, WBC count, RBC count, lymphocytes, monocytes, eosinophils, basophils, band cells, neutrophils, hemoglobin, hematocrit, mean corpuscular volume, mean corpuscular hemoglobin, mean corpuscular hemoglobin concentration, RDW, platelet count, mean platelet volume, CRP, troponin, ferritin, procalcitonin, D-dimer, fibrinogen, PT, INR, aPTT, TT, time to extubation, inpatient days, LNR, and NMR between survival and non-survival groups, and data are expressed as mean ± standard deviation. The chi-squared test was used to compare the prevalence of obesity, T2D, hypertension, coronary heart disease, the number of patients that required IMV, and the sex ratio in survival and non-survival groups, and data are expressed as absolute values or percentages. Receiver operating characteristic (ROC) curves were conducted to calculate area under the curve (AUC) and 95% confidence interval (95% CI) for LNR and NMR. ROC curves together with the Youden index was used to calculate optimal cut-off points, sensitivity, specificity, Relative Risk (RR), Odd Ratio (OR), and 95% CI for LNR and NMR. The sensitivity, specificity, and OR were also calculated for LNR and NMR together by means of the chi-squared test. Multiple regression analysis was performed to estimate the effect of potential confounding variables including obesity, T2D, hypertension, CHD, age, and gender on the accuracy of LNR and NMR as predictors of in-hospital death. Survival analyses were performed for LNR, NMR, and other potential mortality predictors by the Kaplan–Meier method. Differences were considered significant when *p* < 0.05. Statistical analyses were performed by means of the GraphPad Prism 6.01 software (GraphPad Software, La Jolla, CA 92037, USA), the MDCalc Software (New York, NY 10003, USA), the IBM SPSS Statistics version 26.0 (IBM, Armonk, NY, USA), and the R i386 3.5.2 terminal (Microsoft Corp., Boston, MA, USA).

## 3. Results

Figure 1 shows the selection process of patients enrolled in the study based on the inclusion and exclusion criteria.

After hospital admission, 63% (*n* = 34) seriously ill patients with Covid-19 survived, whereas nearly 37% (*n* = 20) patients with a severe course of this disease died (Table 1). Among non-survivors, 75% (*n* = 15) of patients were men (*p* = 0.002). Patients in the non-survival group were, on average, about 9 years older than individuals in the survival group (62.90 ± 14.18 versus 54.06 ± 12.43 years, respectively) (Table 1). In contrast, there were no significant differences between survivors and non-survivors for BMI (28.24 ± 4.60 versus 27.88 ± 4.05 kg/m^2^, respectively). Furthermore, there were no significant differences between survivors and non-survivors with respect to the prevalence of obesity (*p* = 0.277). On the contrary, the prevalence of T2D (*p* = 0.037), hypertension (*p* = 0.006), and CHD (*p* = 0.033) was significantly increased in the non-survival group as compared to the survival group (Table 1). The need for intubation was significantly greater in the non-survival group than in the survival group (83.33 versus 30.43%, respectively) with no significant differences with respect to the time to extubation, number of days that patients spent in hospital, and the five-drug regimen used for treating patients that included aziythromycin, ceftriaxone, enoxaparin sodium, dexamethasone, and acetaminophen (Table 1).

At hospital admission, there were no significant differences between survivors and non-survivors with respect to serum glucose, creatinine, uric acid, triglycerides, total bilirubin, indirect bilirubin, ALT, ALP, total protein, amylase, lipase, CPK, CK-MB, phosphorus, magnesium, sodium, potassium, chlorine, and calcium (Table 2). On the contrary, blood levels of urea (*p* = 0.004), total cholesterol (*p* = 0.010), HDL (*p* = 0.015), LDL (*p* = 0.012), direct bilirubin (*p* = 0.012), AST (*p* = 0.021), GGT (*p* = 0.015), albumin (*p* = 0.001), and LDH (*p* = 0.001) showed statistically significant differences between survivors and non-survivors at hospital admission (Table 2).

There were no significant differences between survivors and non-survivors for prothrombin time, INR, thrombin time, aPTT, fibrinogen, troponin I, and myoglobine (Table 3). On the other hand, non-survivors showed significantly higher values of D-dimer (*p* = 0.017), ferritin (*p* = 0.012), CRP (*p* = 0.014), and procalcitonin (*p* = 0.037) than survivors at the time of admission (Table 3).

No significant differences between survivors and non-survivors were found for most of the hematological parameters including total cell counts of leukocytes, lymphocytes, monocytes, eosinophils, basophils, and RBC (Table 4). However, there were significant differences between survivors and non-survivors with respect to the percentage of neutrophils (*p* = 0.041), lymphocytes (*p* = 0.001) monocytes (*p* = 0.001), and eosinophils (*p* = 0.007), and the neutrophil count (*p* = 0.003) (Table 4).

On average, LNR showed a significant fourfold decrease in non-survivors as compared to survivors (*p* = 0.003) (Figure 2A). In parallel, NMR exhibited a significant twofold increase in non-survivors with respect to survivors (*p* = 0.001) (Figure 2B).

The area under the ROC curve of LNR was 0.832 (95% CI, 0.701–0.922, *p* < 0.001) (Figure 3A). At hospital admission, the best cut-off point for LNR was ≤ 0.088 with a sensitivity of 85.00% and a specificity of 74.19% (Youden index = 0.5919, 95% CI 0.3380–0.7387). The RR and OR for LNR was 5.8933 (95% CI, 1.9661–17.6652) and 16.2917 (95% CI, 3.7550–70.6837), respectively (Figure 3A). LNR was less than 0.088 in 91% of patients with severe Covid-19 that died, whereas only 20% of patients with severe Covid-19 that survived showed LNR ≤ 0.088. Simultaneously, the area under the ROC curve of NMR was 0.890 (95% CI, 0.768–0.962, *p* < 0.001) (Figure 3B). The best cut-off value for NMR was ≥ 17.75 with a sensitivity of 89.47%, specificity of 80.00%, and Youden index of 0.6947 (95% CI, 0.4349–0.8333). The RR and OR for NMR was 8.8542 (95% CI, 2.2864–34.2878) and 27.9286 (95% CI, 5.1435–151.6497), respectively (Figure 3B). NMR was greater than 17.75 in 95% of patients with severe Covid-19 that died, whereas only 10% of patients with severe Covid-19 that survived had NMR ≥ 17.75. Additionally, LNR ≤ 0.088 together with NMR ≥ 17.75 predicted in-hospital mortality in seriously ill patients with Covid-19 with a sensitivity of 81.48%, specificity of 83.33%, and OR of 22.00 (95% CI, 4.556–106.238). The accuracy of LNR and MNR as independent risk factors for in-hospital death was not modified after adjusting by comorbidities, gender, or age in multiple regression analysis.

Table 5 confirmed that LNR and NMR are more accurate to predict mortality in Mexican patients with severe Covid-19 than WBC count by itself. In this sense, the area under the ROC curve of the total leukocyte count was 0.702 (95% CI, 0.557–0.822) (Table 5). The area under the ROC curves for neutrophil, lymphocyte, and monocyte counts were 0.746 (95% CI, 0.605–0.857), 0.735 (95% CI, 0.593–0.849), and 0.605 (95% CI, 0.458–0.739), respectively (Table 5). LNR and MNR were also better predictors for in-hospital mortality than the serum levels of several biochemical and immune parameters. As a matter of fact, the area under the ROC curves for D-dimer, ferritin, CRP, troponin I, LDH, and procalcitonin were 0.730, (95% CI, 0.548–0.869), 0.777 (95% CI, 0.601–0.901), 0.750 (95% CI, 0.569–0.884), 0.656 (95% CI, 0.464–0.816), 0.758 (95% CI, 0.618–0.867), and 0.826 (95% CI, 0.682–0.924), respectively (Table 5).

Survival analysis revealed that patients seriously ill with Covid-19 showing NMR greater than 17.75 at admission have 50% survival probability after 9 days of hospitalization (Figure 4A). Patients with LNR less than 0.088 show 50% survival probability after 12 days of hospitalization (Figure 4B). When using the cut-off points for total leukocytes > 11.4 × 10^3^/μL, neutrophils > 6.3 × 10^3^/μL, lymphocytes ≤ 0.8 × 10^3^/μL, and monocytes ≤ 0.3 × 10^3^/μL separately, the survival probability significantly decrease in patients with severe Covid-19 after 12–20 days of hospitalization (Figure 4C–F, respectively). Despite LDH values significantly differing between survivors and non-survivors (320.63 ± 132.11 versus 475.63 ± 195.83, *p* = 0.001, respectively), patients with LDH > 362 IU/L show a significant decrease in the probability of survival up to 27 days after hospitalization (Figure 4G). In contrast, patients with severe Covid-19 and procalcitonin values greater than 0.1 ng/mL exhibit a 50% decrease in the survival probability after 12 days of hospitalization (Figure 4H).

Taking these results into account, we propose a new triage based on the use of NMR and LNR to assist clinicians to estimate in-hospital death risk in critically ill patients with Covid-19, with the aim of quickly admitting them into intensive care units and reducing the number of fatalities related to this disease (Figure 5).

## 4. Discussion

The search for reliable, sensitive, and specific markers that can help us to discriminate patients with the most severe forms of Covid-19 at the time of hospital admission is still a matter of great urgency. In this sense, the analysis of routine laboratory results has revealed a strongly interdependent relationship among neutrophils, lymphocytes, and monocytes that is linked to the severity of Covid-19 [22].

To date, six retrospective clinical studies examining the potential use of NLR as a prognostic marker for severe Covid-19 have been published; none in Mexican patients until now [22,23,24,25,26,27]. All these clinical studies consistently conclude that NLR is an independent risk factor of mortality in critically ill patients with Covid-19. Concurring with this information, our results indicate that a LNR less than 0.088 at hospital admission is effective to early predict mortality of patients with severe Covid-19 (OR = 16.2917, 95% CI 3.7550–70.6837).

In parallel, we also wanted to explore if the apparent relationship between neutrophil and monocyte counts might be used to evaluate death risk in seriously ill patients with Covid-19. Unexpectedly, our data demonstrate that NMR is even more sensitive and specific than LNR to predict in-hospital mortality of Mexican patients with Covid-19. Indeed, our results demonstrate that having a NMR greater than 17.75 (OR = 27.9286, 95% CI 5.1435–151.6497) at the time of hospital admission is an independent risk factor for in-hospital mortality in patients with severe Covid-19, even after adjusting for comorbidities, gender, and age. Interestingly, the specificity of NMR slightly increases when using together with LNR, although the sensitivity significantly decreases and its possible clinical implementation should be made taking into account this information. Another important aspect that supports the possible implementation of NMR in clinical practice is the fact that this in-hospital mortality marker may assist in the early identification of patients with Covid-19 that will require a more aggressive clinical management. As a matter of fact, survival analyses revealed that using NMR > 17.75 at admission, clinicians might identify patients with severe Covid-19 at higher risk of experiencing 50% reduction in the probability of survival in 9 days. In contrast, identification of patients with poorer survival probability takes longer when using other markers such as lymphocyte count, LDH or procalcitonin. In other words, the use of NMR may help to clinicians to quickly identify patients at higher risk of in-hospital mortality with the aim of giving them priority into intensive care units.

As far as we know, this is the first study showing that NMR can be used as a cheap, fast, and reliable marker of in-hospital mortality in severe Covid-19. Hongmei Zhang and coworkers recently reported that NMR was significantly higher in patients with the most severe forms of Covid-19 as compared to those with mild and moderate courses of the disease [28]. However, these authors also found that NMR failed to predict which patients had higher risk of developing a more severe form of Covid-19 [28]. This apparent controversy in the accuracy of NMR to discriminate patients at higher risk of death might be explained by understanding the impact of ethnicity on the development and clinical presentation of Covid-19. In this sense, several groups mostly from China have proposed that the serum levels of D-dimer, ferritin, CRP, troponin I, LDH, and procalcitonin are good predictors of mortality in patients with severe Covid-19 [8,9,13]. However, our study shows that in Mexican patients these biochemical and immune parameters exhibit poorer ability to predict mortality than that found for LNR and NMR. Again, these controversial findings may be related to differences in the genetic background and prevalence of comorbidities between Mexican-Mestizo subjects of the south-central region of Mexico and East Asian population. For all these reasons, we firmly believe that searching for markers associated with the most severe courses of Covid-19 should be conducted in a population-specific manner. In other words, LNR and NMR seem to be sensitive and specific markers of in-hospital mortality for severe Covid-19 in Mexican patients but no other populations such as Chinese people. Therefore, the clinical use of these ratios in patients with severe Covid-19 that require hospitalization should be carried out with caution in a population-specific manner.

The rationale behind using LNR and NMR to identify patients with severe Covid-19 at higher risk of death is probably given by the differential roles that lymphocytes, monocytes, and neutrophils play during immune response. In humans, neutrophils are the most abundant type of peripheral WBC with prominent functions in the acute inflammatory response by migrating toward injured tissues in response to chemoattractant signals such as interleukin (IL-) 8 [29]. Neutrophils are now considered one of the most important immune cells in defending the airway epithelium against the SARS-CoV-2 infection by locally stimulating the production of IL-1 beta, IL-6, tumor necrosis factor-alpha (TNF-alpha), and reactive oxygen species (ROS) [30]. Paradoxically, neutrophil hyperactivation and recruitment intensify the acute inflammatory response and worsen epithelial tissue damage thus leading to disease progression [30,31]. On the contrary, lymphocytes are very important leukocytes in charge of mediating immune tolerance to self-antigens, activation of pathogen-specific adaptive immunity, and, last but not less important, orchestration of immunomodulatory mechanisms [32,33]. Lymphocytes, including T and B cells, are able to orchestrate immunomodulatory mechanisms via IL-10 and transforming growth factor-beta 1 (TGF-beta 1) production, a couple of cytokines with key anti-inflammatory and wound-healing actions [34]. In parallel, monocytes are circulating leukocytes that in humans can be sorted into three subpopulations based on the cell surface expression of the cluster of differentiation (CD) 14 and CD16 [35]. The classical monocyte subpopulation expresses high CD14 levels with no CD16 expression. Intermediate monocytes show CD14 and CD16 expression; in contrast, non-classical monocytes express very low CD14 levels accompanied by CD16 expression [36]. Intermediate and non-classical monocytes have been shown to produce high levels of IL-1 beta in response to lipopolysaccharide and in patients with metabolic syndrome [37,38]. Interestingly, reduction in both intermediate and non-classical monocyte subpopulations has been recently associated with increased severity of the SARS-CoV-2 infection [39]. Additionally, monocytes can differentiate into alternatively activated macrophages and prevent inflammatory responses by promoting tissue repair via IL-10 and TGF-beta 1 production, which may also play a pivotal role in regulating hyperactivation of the inflammatory response described in patients with severe Covid-19 [40]. In other words, neutrophils and the cell populations of lymphocytes and monocytes seem to have typical antagonistic roles in multiple inflammatory scenarios including that provoked by the SARS-CoV-2 infection. Thus, it is feasible to assume that LNR and NMR might reflect an imbalance among these immune cells that in turn could be related to excessive inflammation and poorer survival in patients with severe Covid-19. For this reason, present results do not only offer novel mortality markers for severe Covid-19 but also reveal potential lines of treatment aimed to decrease neutrophil hyperactivation and increase lymphocyte and monocyte numbers by means of administering drugs as tofacitinib, reparixin, buspirone, and/or lanimostim. However, the possible efficacy of the aforementioned drugs in patients with severe Covid-19 should still be prospectively examined in randomized, placebo-controlled clinical trials.

Finally, concurring with previous literature [41], our study demonstrates that except for obesity, the prevalence of T2D, hypertension, and CHD was higher in the non-survival group than in patients with severe Covid-19 that survived. Likewise, patients in the non-survival group more frequently required IVM than surviving patients. In this way, our results confirm that the presence of previous non-communicable diseases such as T2D, hypertension, and CHD clearly aggravates disease severity and increases the number of fatalities related to Covid-19. So, the SARS-CoV-2 pandemic remarks again the urgency of more effective politics of public health aimed to control the incidence of noncommunicable diseases in countries like Mexico, which is facing a terrible disproportion in the number of deaths related to Covid-19.

This study has some limitations, including the sample size and exclusion of patients with previous diagnosis of other pathologies such as cancer, autoimmune disease, and/or other viral infections such as HIV and HCV. Improvement of these limitations may help to strengthen our findings and extend the use of LNR and NMR to predict in-hospital mortality in patients with high clinical heterogeneity.

## 5. Conclusions

This study demonstrates for the first time that an LNR less than 0.088 and an NMR greater than 17.75 at the time of admission can accurately predict in-hospital mortality of patients with severe Covid-19. Indeed, NMR is significantly more sensitive and specific than LNR to estimate the mortality risk in patients with Covid-19 that meet hospitalization criteria. In Mexican adult patients, both NMR and LNR are better predictors for in-hospital death related to severe Covid-19 than other markers such as D-dimer, ferritin, CRP, troponin I, LDH, and procalcitonin. Together with the presence of T2D, hypertension, CHD, male gender, and increased age (>63 years), NMR and LNR should be considered as independent risk factors for mortality in Mexican patients seriously ill with Covid-19. Simply by dividing the neutrophil count by the monocyte count, or the lymphocyte count by the neutrophil count, at the time of hospital admission, clinicians can have cheap, fast, and reliable predictors for in-hospital death in patients with Covid-19. In practical terms, we propose a new triage based on the use of NMR and LNR to assist clinicians to estimate in-hospital death risk in critically ill patients with Covid-19 with the aim of quickly admitting them into intensive care units and reducing the number of fatalities related to this disease (Figure 5). For this reason, we encourage other clinical research groups to assess the efficacy of these mortality predictors not only in Mexicans but also other Latin-American populations.

## Figures and Tables

**Figure 1 microorganisms-08-01560-f001:**
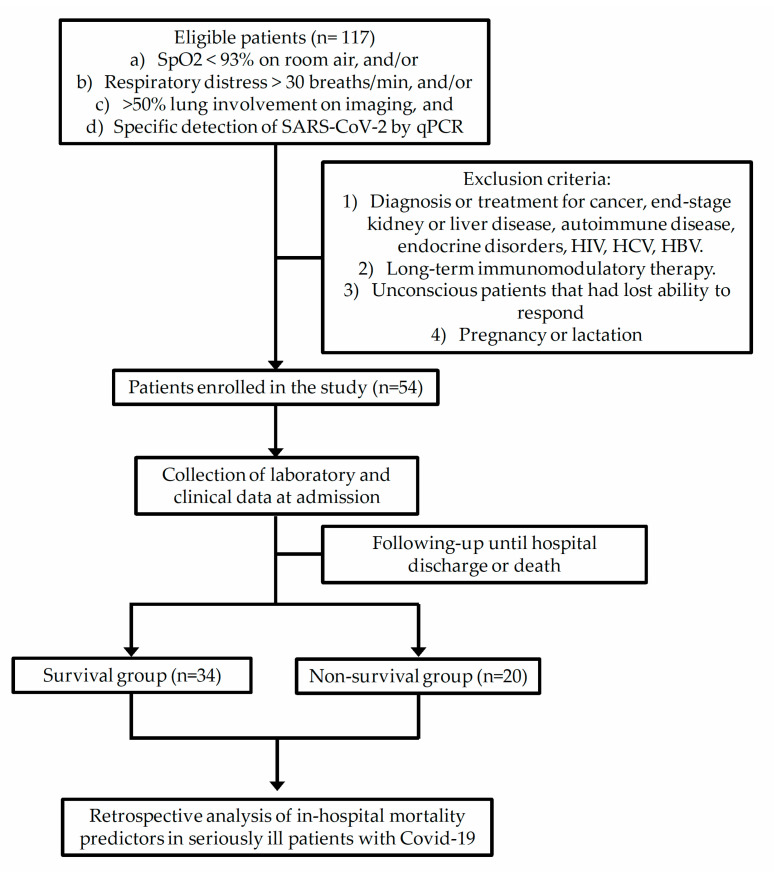
Schematic flow chart showing the selection process of patients enrolled in the study based on the inclusion and exclusion criteria. SpO_2_, oxygen saturation level; SARS-CoV-2, novel severe acute respiratory syndrome coronavirus-2; qPCR, quantitative polymerase chain reaction; HIV, human immunodeficiency virus; HCV, hepatitis C virus; HBV, hepatitis B virus; Covid-19, coronavirus disease 2019.

**Figure 2 microorganisms-08-01560-f002:**
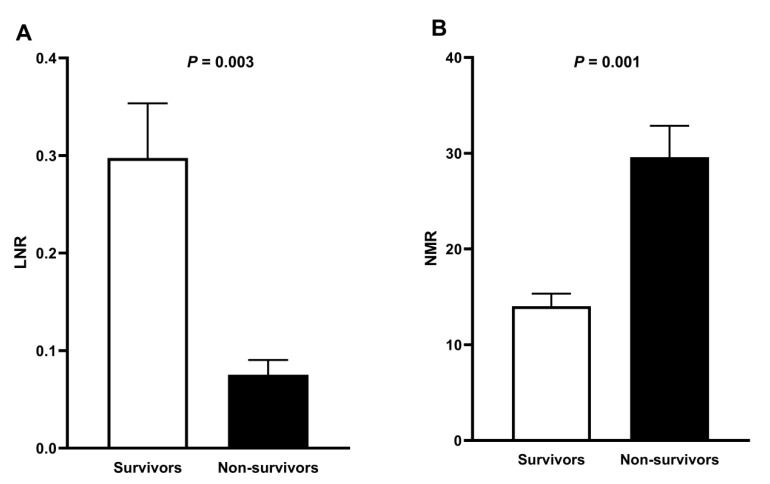
LNR and NMR at admission in hospitalized patients with severe Covid-19. (**A**) LNR showed a significant fourfold decrease in non-survivors as compared to survivors. LNR resulted of dividing the total lymphocyte count by the total neutrophil count. (**B**) NMR exhibited a significant twofold increase in non-survivors with respect to survivors. NMR resulted of dividing the total neutrophil count by the total monocyte count. Normality of data distribution was estimated by the Shapiro–Wilk test. The unpaired Student’s *t*-test was used to compare LNR and MNR between survivors and non-survivors, and data are presented as mean ± standard deviation. Differences were considered significant when *p* < 0.05. LNR, Lymphocyte-to-neutrophil ratio; NMR, neutrophil-to-monocyte ratio.

**Figure 3 microorganisms-08-01560-f003:**
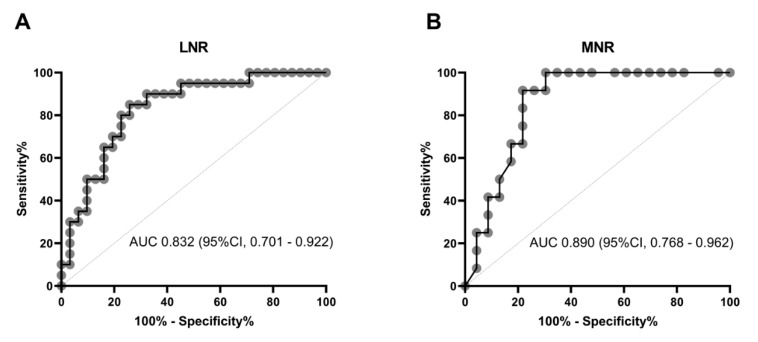
ROC curves for LNR and MNR to predict in-hospital mortality at admission in patients with severe Covid-19. (**A**) For LNR, the best cut-off point to predict in-hospital death was ≤ 0.088 with AUC of 0.832 (95% CI, 0.701–0.922), sensitivity of 85.00%, specificity of 74.19%, Youden index of 0.5919 (95% CI 0.3380–0.7387), RR of 5.8933 (95% CI, 1.9661–17.6652), and OR of 16.2917 (95% CI, 3.7550–70.6837). (**B**) For NMR, the best cut-off point to predict in-hospital death was ≥ 17.75 with AUC of 0.890 (95% CI, 0.768–0.962), sensitivity of 89.47%, specificity of 80.00%, Youden index of 0.6947 (95% CI, 0.4349–0.8333), RR of 8.8542 (95% CI, 2.2864–34.2878), and OR of 27.9286 (95% CI, 5.1435–151.6497). LNR resulted of dividing the total lymphocyte count by the total neutrophil count. NMR resulted of dividing the total neutrophil count by the total monocyte count. Statistical analyses were performed by means of the MDCalc Software (New York, NY 10003, USA) and the IBM SPSS Statistics version 26.0 (IBM, Armonk, NY, USA). Differences were considered significant when *p* < 0.05. LNR, Lymphocyte-to-neutrophil ratio; NMR, neutrophil-to-monocyte ratio; ROC, Receiver Operating Characteristic curves; AUC, area under the ROC curve; CI, confidence interval; RR, relative risk; OR, Odd ratio.

**Figure 4 microorganisms-08-01560-f004:**
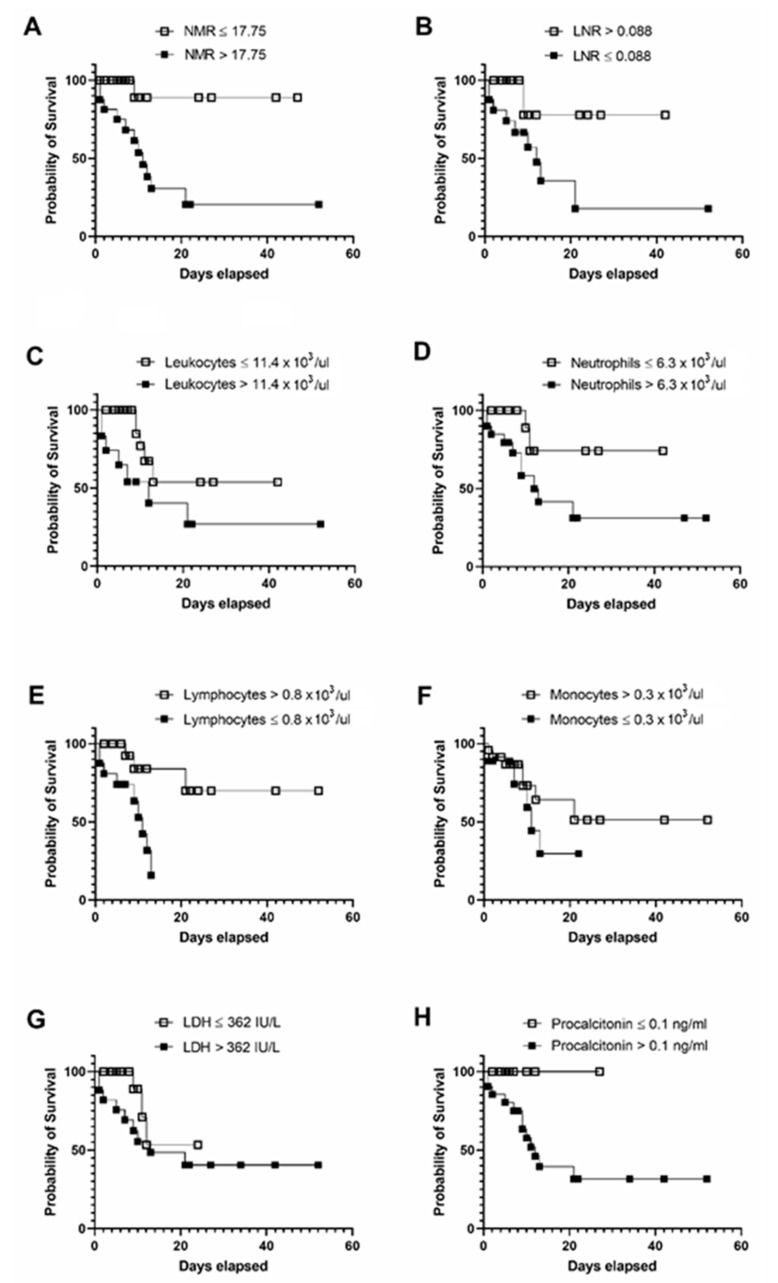
Kaplan–Meier curves for comparing the survival probability of different clinical predictors of in-hospital mortality at admission in patients seriously ill with Covid-19. (**A**) Patients with severe Covid-19 that show NMR > 17.75 at admission show 50% survival probability after 9 days of hospitalization. (**B**) Patients with LNR < 0.088 at admission show 50% survival probability after 12 days of hospitalization. (**C**) Patients with total leukocyte count > 11.4 × 10^3^/μL at admission exhibit a significant decrease in the survival probability up to 20 days after hospitalization. (**D**) Patients with total neutrophil count > 6.3 × 10^3^/μL at admission have 50% survival probability up to 13 days after hospitalization. (**E**) Patients with total lymphocyte count ≤ 0.8 × 10^3^/μL at admission show a significant decrease in the survival probability up to 14 days after hospitalization. (**F**) Patients with total monocyte count ≤ 0.3 × 10^3^/μL at admission exhibit a significant decrease in the survival probability up to 23 days after hospitalization. (**G**) Patients with LDH > 362 IU/L at admission show a significant decrease in the probability of survival up to 27 days after hospitalization. (**H**) Patients with procalcitonin values > 0.1 ng/mL at admission exhibit a 50% decrease in the survival probability after 12 days of hospitalization. Cut-off points for each marker are described at the top of the corresponding panel. Receiver operating characteristic (ROC) curves together with the Youden index was used to calculate optimal cut-off points for all markers. NMR resulted of dividing the total neutrophil count by the total monocyte count. LNR resulted of dividing the total lymphocyte count by the total neutrophil count. Survival analyses were performed by means of the Kaplan–Meier method using the GraphPad Prism 6.01 software. NMR, neutrophil-to-monocyte ratio; LNR, Lymphocyte-to-neutrophil ratio; LDH, lactate dehydrogenase.

**Figure 5 microorganisms-08-01560-f005:**
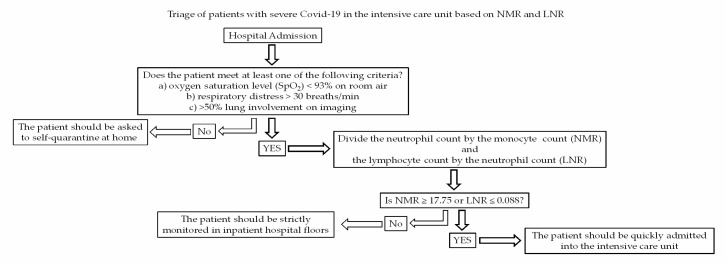
Proposal of triage based on NMR and LNR for patients seriously ill with Covid-19 that meet hospitalization criteria. At hospital admission, the neutrophil count should be divided by the monocyte count (NMR) and the lymphocyte count by the neutrophil count (LNR). If NMR ≥ 17.75 or LNR ≤ 0.088, the patient should be quickly admitted into the intensive care unit.

**Table 1 microorganisms-08-01560-t001:** Demographic and clinical characteristics in hospitalized patients with severe Covid-19. Abbreviations: W, women; M, men; BMI, body mass index; T2D, type 2 diabetes; IMV, invasive mechanical ventilation. Normality of data distribution was estimated by the Shapiro–Wilk test. The unpaired Student’s T-test was used to compare numerical variables and data are presented as mean ± standard deviation. The chi-squared test was used to compare categorical variables and data are expressed as absolute values or percentages. * Differences were considered significant when *p* < 0.05.

Parameters	Survivors (*n* = 34)	Non-Survivors (*n* = 20)	*p* Value
Gender (W/M)	21/13	5/15	0.002 *
Age (years)	54.06 ± 12.43	62.9 ± 14.18	0.020 *
BMI (kg/m^2^)	28.24 ± 4.60	27.88 ± 4.05	0.903
Obesity prevalence (%)	52.17	41.66	0.277
T2D prevalence (%)	43.47	75.00	0.037 *
Hypertension prevalence (%)	17.39	58.33	0.006 *
Coronary heart disease (%)	8.69	33.33	0.033 *
IMV (%)	30.43	83.33	0.001 *
Time to extubation (days)	2.43 ± 0.79	3.66 ± 0.82	0.167
Inpatient days (days)	15.65 ± 3.13	8.41 ± 1.66	0.060
Drug regimen	Aziythromycin, ceftriaxone, enoxaparin sodium, dexamethasone, and acetaminophen		-

**Table 2 microorganisms-08-01560-t002:** Biochemical parameters, kidney function tests, and liver function tests at admission in hospitalized patients with severe Covid-19. Abbreviations: HDL, high-density lipoproteins; LDL, low-density lipoproteins; ALT, alanine aminotransferase; AST, aspartate aminotransferase; ALP, alkaline phosphatase; GGT, gamma glutamyl transferase; LDH, lactate dehydrogenase; CPK, creatinine phosphokinase; CK-MB, creatinine kinase myocardial band. Normality of data distribution was estimated by the Shapiro–Wilk test. The unpaired Student’s *t*-test was used to compare numerical variables and data are presented as mean ± standard deviation. * Differences were considered significant when *p* < 0.05.

Parameters	Survivors (*n* = 34)	Non-Survivors (*n* = 20)	*p* Value
Glucose (mg/dL)	148.19 ± 92.13	148.16 ± 60.76	0.999
Urea (mg/dL)	42.05 ± 37.06	91.07 ± 77.16	0.004 *
Creatinine (mg/dL)	0.995 ± 1.18	1.85 ± 2.79	0.133
Uric Acid (mg/dL)	5.65 ± 2.81	7.39 ± 4.09	0.097
Total Cholesterol (mg/dL)	151.77 ± 34.98	126.06 ± 20.85	0.010 *
Triglycerides (mg/dL)	166.04 ± 67.93	169.75 ± 54.89	0.853
HDL (mg/dL)	34.30 ± 10.88	23.92 ± 12.09	0.015 *
LDL (mg/dL)	95.30 ± 30.14	70.38 ± 18.69	0.012 *
Total bilirubin (mg/dL)	0.684 ± 0.401	0.804 ± 0.315	0.349
Direct bilirubin (mg/dL)	0.185 ± 0.133	0.323 ± 0.215	0.012 *
Indirect bilirubin (mg/dL)	0.492 ± 0.254	0.498 ± 0.153	0.939
ALT (IU/L)	35.42 ± 26.29	41.47 ± 28.92	0.465
AST (IU/L)	34.74 ± 22.86	59 ± 47.74	0.021 *
ALP (IU/L)	91.00 ± 27.22	125.29 ± 120.86	0.134
GGT (IU/L)	69.54 ± 43.07	124.33 ± 101.26	0.015 *
Total Protein (mg/dL)	6.59 ± 0.525	6.31 ± 0.627	0.099
Albumin (mg/dL)	3.59 ± 0.479	2.94 ± 0.409	0.001 *
LDH (IU/L)	320.63 ± 132.11	475.63 ± 195.83	0.001 *
Amylase (IU/L)	46.1 ± 35.39	56.83 ± 26.93	0.429
Lipase (IU/L)	116.00 ± 304.00	52.36 ± 52.27	0.502
CPK (IU/L)	101.52 ± 97.49	699.57 ± 1937.82	0.114
CK-MB (IU/L)	23.80 ± 11.90	44.43 ± 59.39	0.099
Phosphorus (mg/dL)	3.89 ± 2.01	4.25 ± 1.67	0.516
Magnesium (mg/dL)	2.74 ± 3.53	2.42 ± 0.629	0.698
Sodium (mEq/L)	136.00 ± 6.56	138.53 ± 6.31	0.182
Potassium (mEq/L)	5.56 ± 6.67	4.47 ± 0.719	0.484
Chlorine (mEq/L)	100.28 ± 7.19	101.84 ± 6.49	0.441
Calcium (mg/dL)	8.63 ± 0.686	8.14 ± 1.20	0.076

**Table 3 microorganisms-08-01560-t003:** Coagulation and immune parameters at admission in hospitalized patients with severe Covid-19. Abbreviations: INR, international normalized ratio; aPTT, activated partial thromboplastin time; CRP, C-reactive protein. Normality of data distribution was estimated by the Shapiro–Wilk test. The unpaired Student’s *t*-test was used to compare numerical variables and data are presented as mean ± standard deviation. * Differences were considered significant when *p* < 0.05.

Parameters	Survivors (*n* = 34)	Non-Survivors (*n* = 20)	*p* Value
Prothrombine time (s)	11.97 ± 2.49	12.96 ± 1.55	0.237
INR	0.995 ± 0.246	1.10 ± 0.167	0.216
Thrombin time (s)	16.76 ± 1.61	17.78 ± 1.42	0.085
aPTT (s)	26.18 ± 6.99	26.36 ± 5.68	0.941
Fibrinogen (mg/dL)	637.23 ± 222.29	703.75 ± 206.92	0.400
D-dimer (Ug/L)	975.33 ± 477.81	8260.33 ± 13354.39	0.017 *
Ferritin (ng/mL)	522.62 ± 451.93	937.00 ± 415.09	0.012 *
CRP (mg/L)	129.43 ± 90.10	221.07 ± 108.32	0.014 *
Troponin I (ng/mL)	37.68 ± 63.57	68.458 ± 112.73	0.338
Myoglobine (ng/L)	94.49 ± 107.76	254.38 ± 353.69	0.089
Procalcitonin (ng/mL)	0.220 ± 0.256	2.01 ± 4.07	0.037 *

**Table 4 microorganisms-08-01560-t004:** Hematological parameters at admission in hospitalized patients with severe Covid-19. Abbreviations: MCV, mean corpuscular volume; MHC, mean corpuscular hemoglobin; RDW, red blood cell distribution. Normality of data distribution was estimated by the Shapiro–Wilk test. The unpaired Student’s *t*-test was used to compare numerical variables and data are presented as mean ± standard deviation. * Differences were considered significant when *p* < 0.05.

Parameters	Survivors (*n* = 34)	Non-Survivors (*n* = 20)	*p* Value
Leukocytes (×10^3^/μL)	13.48 ± 25.44	13.19 ± 6.34	0.960
Neutrophil percentage (%)	73.95 ± 17.19	84.72 ± 18.89	0.041 *
Lymphocyte percentage (%)	16.79 ± 11.35	7.48 ± 5.65	0.001 *
Monocyte percentage (%)	5.96 ± 2.54	3.35 ± 1.42	0.001 *
Band cells (%)	0.000 ± 0.000	0.08 ± 0.358	0.217
Eosinophil percentage (%)	0.739 ± 0.974	0.110 ± 0.281	0.007 *
Basophil percentage (%)	0.429 ± 1.23	0.085 ± 0.123	0.221
Neutrophils (×10^3^/μL)	7.39 ± 4.44	11.93 ± 5.99	0.003 *
Lymphocytes (×10^3^/μL)	2.09 ± 4.40	0.750 ± 0.426	0.181
Monocytes (×10^3^/μL)	0.519 ± 0.256	0.450 ± 0.276	0.364
Eosinophils (×10^3^/μL)	0.096 ± 0.272	0.015 ± 0.049	0.193
Basophils (×10^3^/μL)	0.339 ± 1.87	0.000 ± 0.000	0.423
Erythrocyte (×10^6^/μL)	4.71 ± 0.893	4.75 ± 1.09	0.884
Hemoglobin (g/dL)	14.17 ± 2.56	14.42 ± 3.11	0.756
Hematocrit (%)	42.40 ± 7.58	42.82 ± 9.44	0.863
MCV (fL)	90.54 ± 5.88	91.35 ± 4.15	0.597
MCH (pg)	30.34 ± 2.73	30.42 ± 1.64	0.905
RDW (%)	15.07 ± 3.38	14.72 ± 2.07	0.685
Platelets (×10^3^/μL)	266.61 ± 111.11	240.25 ± 113.83	0.416

**Table 5 microorganisms-08-01560-t005:** Area under the ROC curves for hematological, biochemical, and immune parameters to predict in-hospital mortality at admission in patients with severe Covid-19. Statistical analyses were performed by means of the MDCalc Software (New York, NY 10003, USA) and the IBM SPSS Statistics version 26.0 (IBM, Armonk, NY, USA). Differences were considered significant when *p* < 0.05. Abbreviations: CRP, C-reactive protein; LDH, lactate dehydrogenase; ROC, Receiver Operating Characteristic curves; AUC, area under the ROC curve; CI, confidence interval.

Parameters	AUC	CI 95%
Total leukocyte count	0.702	0.557–0.822
Neutrophil count	0.746	0.605–0.857
Lymphocyte count	0.735	0.593–0.849
Monocyte count	0.605	0.458–0.739
D-dimer	0.730	0.548–0.869
Ferritin	0.777	0.601–0.901
CRP	0.750	0.569–0.884
Troponin I	0.656	0.464–0.816
LDH	0.758	0.618–0.867
Procalcitonin	0.826	0.682–0.924

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
