# Peer review of "The Neutrophil-to-Monocyte Ratio and Lymphocyte-to-Neutrophil Ratio at Admission Predict In-Hospital Mortality in Mexican Patients with Severe SARS-CoV-2 Infection (Covid-19)"

_microorganisms, 2020, doi:10.3390/microorganisms8101560_

Round 1
Reviewer 1 Report
The authors presented a manuscript aimed at proposing a novel triage of suspected COVID-19 patients based on the evaluation of NMR and LNR. In particular, the authors analyzed a case series of 54 COVID-19 patients who survived or died as a result of the infection. For each patient, the authors analyzed the levels of several hematological and immunological biomarkers including, lymphocyte count, monocyte count, neutrophil count, d-dimer, fibrin, etc. As results the authors obtained that the evaluation of NMR and LNR, together with the analysis of other potential biomarkers, could be predictive for patients' survival. Overall, the authors presented interesting results, however, there are some issues that they have to address before publication:
1) In the abstract section, the authors state “Twenty patients with severe Covid-19 died and 3 in 4 were men older
than 62.90±14.18 years on average.”. With “3 in 4” did the authors mean 75% of patients? If yes, please indicate the correct percentage;
2) Have the authors evaluated the sensitivity and specificity of the concomitant evaluation of both LNR and NMR? Please add this fundamental analysis;
3) In section “2.1. Subjects”, please schematically report the inclusion and exclusión criteria used;
4) Why did the authors evaluate the lymphocyte/neutrophil ratio instead of the neutrophil/lymphocyte ratio? In similar studies, the NLR is generally evaluated. Please argue this choice and see the following manuscript:
– 10.1016/j.ijantimicag.2020.106017
– 10.1017/S0950268820001557
– 10.1016/j.intimp.2020.106504
5) It would also have been interested to evaluate the serum levels of inflammatory cytokines including IL-6, TNF-a, IL10, IL-1B etc, as cytokine storm is one of the key features of COVID-19 infection and it could be strictly associated to patients' survival. Have the authors collected data about cytokine expression?;
6) If authors have collected also survival data, it would be interesting to perform Kaplan-Meier analyses taking into account the baseline levels of NMR, LNR (or NLR), and other relevant markers here identified. What is the authors’ opinion?;
7) It would have been interested to evaluate the predictive value of LNR and NMR in patients with innate or acquired immunodeficiency, such as cancer patients. Did the authors collect data on this? In the Discussion section, the authors should mention the immunological status of these fragile patients positive to COVID-19 and mention the use of immunotherapy for the treatment of both cancer patients and other patients with COVID-19 infection. In this context, the immunomodulatory action of immune checkpoint inhibitors could positively modulate both NRM and LNR with beneficial effects for the patient with severe symptoms. For this purpose, see:
– 10.3390/cancers12082237
– 10.1016/j.critrevonc.2020.103059
– 10.1016/j.lfs.2020.118167
8) The manuscript needs some minor English revisions (e.g. “Patients were excluded of the study if they..” substitute with “Patients were excluded from the study if they…”; etc.).
Author Response
REVIEWER #1
The authors presented a manuscript aimed at proposing a novel triage of suspected COVID-19 patients based on the evaluation of NMR and LNR. In particular, the authors analyzed a case series of 54 COVID-19 patients who survived or died as a result of the infection. For each patient, the authors analyzed the levels of several hematological and immunological biomarkers including, lymphocyte count, monocyte count, neutrophil count, d-dimer, fibrin, etc. As results the authors obtained that the evaluation of NMR and LNR, together with the analysis of other potential biomarkers, could be predictive for patients' survival. Overall, the authors presented interesting results, however, there are some issues that they have to address before publication:
Reply (R)
We want to thank to the Reviewer for her/his very kind comments regarding our work. All your questions, comments, and concerns have indubitably improved the last version of this manuscript.
Query (Q) 1
1) In the abstract section, the authors state “Twenty patients with severe Covid-19 died and 3 in 4 were men older
than 62.90±14.18 years on average.”. With “3 in 4” did the authors mean 75% of patients? If yes, please indicate the correct percentage;
R1
Following the Reviewer’s suggestion, we have indicated the proportion of men in the non-survival group as percentage. Please find this change marked with yellow color at page 1, line 40.
Q2
2) Have the authors evaluated the sensitivity and specificity of the concomitant evaluation of both LNR and NMR? Please add this fundamental analysis;
R2
Following the accurate Reviewer’s suggestion, we have calculated the sensitivity and specificity of both LNR and NMR for predicting mortality in patients with Covid-19. As a matter of fact, LNR ≤ 0.088 together with NMR ≥ 17.75 predicts in-hospital mortality with a sensitivity of 81.48%, specificity of 83.33%, and Odd Ratio (OR) of 22.00 (95% CI, 4.556-106.238). Despite the sensitivity of LNR+NMR is lower than that observed for LNR and MNR separately, the specificity value does increase and we have decided to include this finding in the manuscript. Please find this information marked with green color at pages 4, 10, 11 and 15.
We want to thank you for your very constructive feedback that has indubitably improved the last version of our work.
Q3
3) In section “2.1. Subjects”, please schematically report the inclusion and exclusión criteria used;
R3
Following the Reviewer’s suggestion, we have included a schematic flow chart showing the selection process of patients enrolled in the study based on the inclusion and exclusion criteria. Please find this new information marked with grey color at pages 4 and 5, and in the new figure 1. Thank you for your suggestion. We think the selection process of patients is now clearer.
Q4
4) Why did the authors evaluate the lymphocyte/neutrophil ratio instead of the neutrophil/lymphocyte ratio? In similar studies, the NLR is generally evaluated. Please argue this choice and see the following manuscript:
– 10.1016/j.ijantimicag.2020.106017
– 10.1017/S0950268820001557
– 10.1016/j.intimp.2020.106504
R4
Thank you for your clever observation. We decided to use LNR instead of NLR because we frequently found patients with Covid-19 that show no lymphocyte counts (three in fifty-four, in our case). In other words, there are some Covid-19 patients with lymphocyte counts equal to zero and even though the patient is a real clinical case, it is impossible to calculate a ratio where the denominator is equal to zero. Therefore, calculation of LNR provides the opportunity to estimate in-hospital death risk in all patients no matter they may have no lymphocyte counts reported in laboratory findings.
Q5
5) It would also have been interested to evaluate the serum levels of inflammatory cytokines including IL-6, TNF-a, IL10, IL-1B etc, as cytokine storm is one of the key features of COVID-19 infection and it could be strictly associated to patients' survival. Have the authors collected data about cytokine expression?;
R5
Thank you for your very nice suggestion. We are already working on measuring the levels of proinflammatory and anti-inflammatory cytokines such as TNF-alpha, IL-1 beta, IL-6, and IL-10, among others, in both serum and bronchoalveolar lavage fluid with the aim of characterizing the cytokine storm profile in seriously and critically ill patients with Covid-19. However, it is still an ongoing work that we hope to report in a future communication.
Q6
6) If authors have collected also survival data, it would be interesting to perform Kaplan-Meier analyses taking into account the baseline levels of NMR, LNR (or NLR), and other relevant markers here identified. What is the authors’ opinion?;
R6
Following the accurate Reviewer’s suggestion, we have performed survival analysis by the Kaplan-Meier method for NMR and LNR, as well for the markers that showed the most significant changes between survivors and non-survivors with good values of AUC, sensitivity, and specificity including total leukocyte count, neutrophil count, lymphocyte count, monocyte count, LDH, and procalcitonin. The suggestion of these analyses revealed that using NMR, clinicians might quickly identify to patients with severe Covid-19 at risk of experiencing 50% reduction in the probability of survival in the next nine days with the aim of giving them priority into the intensive care units. For this reason, we want to thank to the Reviewer for her/his very clever insight. Please find this information marked with red color at pages 4, 12, 13 and 15, and in the new figure 4.
Q7
7) It would have been interested to evaluate the predictive value of LNR and NMR in patients with innate or acquired immunodeficiency, such as cancer patients. Did the authors collect data on this? In the Discussion section, the authors should mention the immunological status of these fragile patients positive to COVID-19 and mention the use of immunotherapy for the treatment of both cancer patients and other patients with COVID-19 infection. In this context, the immunomodulatory action of immune checkpoint inhibitors could positively modulate both NRM and LNR with beneficial effects for the patient with severe symptoms. For this purpose, see:
– 10.3390/cancers12082237
– 10.1016/j.critrevonc.2020.103059
– 10.1016/j.lfs.2020.118167
R7
We think the Reviewer’s observation is of enormous scientific and clinical soundness. Unfortunately, we decided not to enroll patients with innate or acquired immunodeficiency in this study, including patients with any type of cancer and HIV-seropositive subjects or having developed AIDS. The rationale behind this decision was in fact the use of immunotherapy for the treatment of these entities that (we believe) does not only has an impact on NMR and LNR but also the entire immune response against SARS-CoV-2 infection. In other words, in this study we only enrolled patients with no immune disease or treatment that together could compromise the ability of the immune system to respond to the SARS-CoV-2. We agree this is a limitation of the study because in real life patients unfortunately get cancer, autoimmune disease, and/or HIV at the same time that Covid-19, and NMR and LNR do not necessarily provide useful information regarding in-hospital mortality risk in this kind of patients. In fact, we included this information at the end of the discussion section and you will be able to find it marked with yellow color at page 17. We hope we can extend our findings to patients with innate or acquired immunodeficiency by conducting another clinical study in the near future.
Q8
8) The manuscript needs some minor English revisions (e.g. “Patients were excluded of the study if they..” substitute with “Patients were excluded from the study if they…”; etc.).
R8
Following the Reviewer’s observation, we have revised the English grammar of the manuscript and corrected as suggested. Thank you. Please find these changes all along the main text. With regard to the abovementioned observation, please find these specific changes marked with pink color at page 3.

Reviewer 2 Report
The work presented by Rizo-Téllez and collaborators reaches the scope of the research journal. The article is well written and easy to read, the study is well conducted, and the conclusions are fully supported by results. I have only a minor comment about the description of monocytes’ role in inflammation in the discussion, that is much more complex than the differentiation in alternatively activated macrophages. I would include a better explanation of their functions, also considering the different monocyte subpopulations and a speculation of how this is related with a stronger immune/inflammatory reaction against SARS-CoV-2.
Author Response
REVIEWER #2
The work presented by Rizo-Téllez and collaborators reaches the scope of the research journal. The article is well written and easy to read, the study is well conducted, and the conclusions are fully supported by results. I have only a minor comment about the description of monocytes’ role in inflammation in the discussion, that is much more complex than the differentiation in alternatively activated macrophages. I would include a better explanation of their functions, also considering the different monocyte subpopulations and a speculation of how this is related with a stronger immune/inflammatory reaction against SARS-CoV-2.
Replay
We want to thank to the Reviewer for her/his very kind comments on our work. Following the Reviewer’s suggestion, we have added a much more extensive description of human monocyte subpopulations with special emphasis on the SARS-CoV-2 infection. We believe the Reviewer’s comment has improved our discussion regarding the possible role of monocytes in Covid-19. Please find this new information marked with pink color at page 16.
